# Theory of MOCVD Growth of III-V Nanowires on Patterned Substrates

**DOI:** 10.3390/nano12152632

**Published:** 2022-07-30

**Authors:** Vladimir G. Dubrovskii

**Affiliations:** Faculty of Physics, St. Petersburg State University, Universitetskaya Emb. 13B, 199034 St. Petersburg, Russia; dubrovskii@mail.ioffe.ru

**Keywords:** MOCVD growth, patterned substrates, additional material flux, desorption, modeling

## Abstract

An analytic model for III-V nanowire growth by metal organic chemical vapor deposition (MOCVD) in regular arrays on patterned substrates is presented. The model accounts for some new features that, to the author’s knowledge, have not yet been considered. It is shown that MOCVD growth is influenced by an additional current into the nanowires originating from group III atoms reflected from an inert substrate and the upper limit for the group III current per nanowire given by the total group III flow and the array pitch. The model fits the data on the growth kinetics of Au-catalyzed and catalyst-free III-V nanowires quite well and should be useful for understanding and controlling the MOCVD nanowire growth in general.

## 1. Introduction

III-V semiconductor nanowires (NWs) are promising building blocks for fundamental studies in nanoscience and nanotechnology [1] and for applications in nanophotonic devices, including those monolithically integrated with the Si electronic platform [2,3,4]. Very efficient relaxation of elastic stress on free NW sidewalls enables the dislocation-free growth of lattice-mismatched NW heterostructures as well as NW growth on Si substrates [5,6,7,8,9,10]. Most III-V NWs are obtained using the vapor–liquid–solid (VLS) method [11] with metal catalysts, often Au [6,9,10,11,12,13,14,15,16], or catalyst-free selective area epitaxy (SAE) [17]. In some cases, the catalyst-free SAE and catalytic VLS III-V NWs can even coexist in one sample depending on the NW radius and growth conditions [18]. MOCVD, or metal-organic vapor phase epitaxy, is the standard technique for the fabrication of large area, high quality ensembles of III-V NWs [13,14,15,16,17,18]. MOCVD growth of III-V NWs on patterned substrates with regular positioning of Au nanoparticles [4,14,15,17,18] enables one to fabricate more uniform NW arrays without any parasitic structures on a surface between the NWs, which is highly desired for applications. The NW growth selectivity is ensured by an inert mask (such as SiO_x_) reflecting a group III precursor, or simply by the absence of precursor decomposition on a substrate surface. 

Most models for MOCVD growth of III-V NWs [19,20,21,22,23,24] consider surface diffusion of group III adatoms as the governing mechanism of material exchange between the substrate surface and NWs, similar to Ref. [25] for Si NWs or Refs. [26,27] for VLS III-V NWs grown by molecular beam epitaxy (MBE). A detailed review of these growth models can be found, for example, in Ref. [28]. An important step in understanding the MBE growth of III-V NWs on a masked surface has been taken in Ref. [29]. It has been shown that Ga-catalyzed GaP NWs fabricated in regular arras of patterned pinholes in the SiO_2_ mask on Si(111) grow from the direct vapor flux plus the additional flux of Ga atoms re-emitted from the mask rather than Ga adatoms diffusing from the mask. The re-emitted Ga flux cancels only when no Ga atoms can impinge the mask surface due to the shadowing effect [29,30,31] in the directional MBE method. An analytic growth model for re-emitted growth species has been developed in Ref. [32], and the differences between the re-emission and surface diffusion mechanisms of material exchange between the substrate and NWs have been further discussed in Refs. [33,34]. 

Here, I try to develop a similar approach for MOCVD growth. I speculate that, in the SAE-MOCVD growth of III-V NWs, the growth species may not adsorb on the substrate surface because no growth normally occurs between the NWs. This would not be the case if group III atoms were able to land and freely diffuse on a substrate. Rather, group III species should desorb from the substrate and then contribute into the NW growth. This process is somewhat similar to the Ga re-emission considered in Refs. [29,32] for MBE growth. On the other hand, the total material current into each NW should be limited by a maximum flux which depends on the pitch of the NW array [24] even in the absence of the shadowing effect. Using these reasonable assumptions, the analytic model provides a self-consistent description of NW growth by MOCVD in different stages and fits the data quite well on the Au-catalyzed VLS InAs NWs [14,20] and catalyst-free GaAs NWs [17] grown by MOCVD on patterned substrates. Self-limiting MOCVD growth influenced by re-emitted group III atoms is discussed here for the first time to my knowledge. It gives a refined picture of the NW growth kinetics and explains the separation between the known growth stages (such as exponential and linear growths [14] yielding different radius dependences of the NW length [14,20,21]) from a different perspective. This approach is quite general and should be useful for understanding and tuning the morphology of MOCVD-grown NWs in a wide range of material systems. 

## 2. Model

I consider an array of identical cylindrical NWs that have the same radius R and length *L* above the substrate surface, neglecting the size inhomogeneity in the first approximation. It is known that the length distributions of Ga-catalyzed GaAs NWs grown by MBE can be very narrow (sub-Poissonian) [31]. For InAs NWs grown in regular arrays by Au-catalyzed MOCVD with the axial growth rates scaling linearly with length, the length distributions are broader [14]. However, the ratio of the length distribution width over the mean length remains much smaller than unity [14]. The contact angle of the catalyst droplets resting on the tops of VLS NWs equals β. Flat tops of catalyst-free SAE NWs correspond to β=0. NWs are grown by selective area MOCVD in the regular array of pitch P without any parasitic growth between the NWs. The surface area per NW equals $kP^{2}$, where k is a geometrical constant (for example, *k* = 1 for square array and $k=\sqrt{3}/2$ for hexagonal array of NWs). The Au-catalyzed VLS or SAE growth of III-V NWs is controlled by the kinetics of group III species under group V-rich conditions [14,19,20,21,22,23,28]. Therefore, the central object for the NW growth kinetics is the total volumetric current of group III atoms into the NW $F_{tot}$ (measured in nm^3^/s). This current is used to increase the NW volume and possibly the droplet volume on the NW top according to [32,33,34]
(1)$$F_{tot}=\frac{d}{dt}[\pi R^{2}L+\frac{\Omega_{35}}{\Omega_{3}}\frac{\pi R^{3}}{3}f\left(\beta \right)]$$

Here, Ω_35_ is the elementary volume per III-V pair in solid, $\Omega_{3}$ is the elementary volume per group III atom in liquid, and f(β)=(1−cosβ)(2+cosβ)/[(1+cosβ)sinβ] is the geometrical function of the droplet contact angle (f(β)=0 for catalyst-free SAE NWs). 

In the case of MBE growth, long enough NWs are able to adsorb all arriving group III atoms, which is why the maximum group III current per NW is given by $F_{max}=v_{3}kP^{2}$, with $v_{3}$ as the incoming flux of group III atoms in nm/s [29,32]. In the case of MOCVD growth, which is closer to equilibrium compared to MBE, a fraction of group III atoms should always desorb from the surface. Therefore, I assume that $F_{max}=v_{3}cP^{2}$, with *c* = *kχ_s_* and $\chi_{s}<1$, is the effective adsorption coefficient. The direct vapor flux into each NW is given by $F_{dir}=v_{3}S_{nw}$, with $S_{nw}$ being the effective surface area of the NW, including the droplet [14,18,19,20,21,22,23,24]. The current $F_{mis}=F_{max}-F_{dir}=v_{3}(cP^{2}-S_{nw})$ lands on the substrate surface rather than on the NW surface. I further assume that a fraction of this missed flux can re-emit from the substrate, land on the NW surface, and also contribute into the NW growth. The additional current into the NW is then given by $F_{add}=F_{mis}S_{nw}/cP^{2}$, similarly to Ref. [32]. The total current into the NW equals $F_{tot}=F_{dir}+F_{add}$. One can thus write
(2)$$F_{tot}=v_{3}S_{nw}(2-\frac{S_{nw}}{cP^{2}}),S_{nw}\leq \min(cP^{2},{\tilde{S}}_{nw}),$$
where ${\tilde{S}}_{nw}$
is the effective collection area of group III atoms for NWs whose length is larger than the desorption-limited diffusion length of group III adatoms on the NW sidewalls. Here, I take into account that the total current cannot be greater than $cP^{2}$, and can be limited by the desorption of group III species from the NW sidewalls and droplet surface. Therefore, the effective surface areas in Equation (2) are given by [14,18,19,20,21,22,23,24,28]
(3)$$S_{nw}=\chi_{d}\frac{2\pi R^{2}}{1+cos\beta }+\chi_{nw}2\pi RL,\;L\leq \lambda_{3}^{des},$$
(4)$${\tilde{S}}_{nw}=\chi_{d}\frac{2\pi R^{2}}{1+cos\beta }+\chi_{nw}2\pi R\lambda_{3}^{des},\;L\geq \lambda_{3}^{des}.$$

Here, $\chi_{d}$ is the pyrolysis efficiency of the group III precursor on the droplet surface, $\chi_{nw}$ is the pyrolysis efficiency on the NW sidewalls, and $\lambda_{3}^{des}$ is the desorption-limited diffusion length of group III adatoms on the NW sidewalls. The first term in Equations (3) and (4) stands for group III atoms collected by the droplet surface or NW top facet for catalyst-free SAE NWs. The second terms in Equations (3) and (4) describe the diffusion current of group III atoms collected by the NW sidewalls, which the maximum collection length limited by $\lambda_{3}^{des}$.

In dilute NW arrays with ${\tilde{S}}_{nw}<cP^{2}$, the total current of group III atoms into the NW is limited by desorption. For long enough NWs, the total current becomes
(5)$$F_{tot}=v_{3}{\tilde{S}}_{nw}(2-\frac{{\tilde{S}}_{nw}}{cP^{2}}),\;S_{nw}>{\tilde{S}}_{nw},$$
with ${\tilde{S}}_{nw}$ given by Equation (4). In dense NW arrays with ${\tilde{S}}_{nw}>cP^{2}$, the total current is limited by its maximum value determined by the group III flow onto the surface and the pitch of the NW array. For long enough NWs, the total current becomes
(6)$$F_{tot}=v_{3}cP^{2},\;S_{nw}>cP^{2}.$$

I will now treat the NW morphological evolution governed by the group III currents given by Equations (2), (5), or (6) in different stages of growth. 

## 3. Results and Discussion

In self-catalyzed VLS growth of GaAs and other III-V NWs (usually, by MBE [35]), Ga droplet serves as a non-stationary reservoir of Ga which may either swell or shrink during growth depending on the V/III flux ratio [29,33], or even self-equilibrate to a certain stationary size [36,37]. In the Au-catalyzed VLS growth of III-V NWs, Au-rich alloy droplets can stabilize the droplet volume. Therefore, Au-catalyzed VLS growth often proceeds at a constant NW radius ($R=R_{0}=const$) and contact angle (β=const) [8,19,20,21,22,26,34]. In this case, Equations (1) and (2) give the axial growth rate of short NWs of the form
(7)$$\frac{dy}{dx}=y\left(2-y\right),\;$$
with $y=S_{nw}/cP^{2}\leq 1$
, and $x=2\chi_{nw}H/R_{0}$, where $H=v_{3}t$ is the effective deposition thickness at a given deposition rate $v_{3}$. Solving Equation (7) with the initial condition L(H=0)=0, the NW length is obtained in the following form:(8)$$L=\frac{\chi_{d}}{\chi_{nw}}\frac{R_{0}}{\left(1+cos\beta \right)}\left[\frac{2}{a+\left(2-a\right)exp\left(-\frac{4\chi_{nw}H}{R_{0}}\right)}-1\right],\;H\leq H_{\lambda },\;a=\frac{\chi_{d}2\pi R_{0}^{2}}{\left(1+cos\beta \right)cP^{2}}.$$
with $H_{\lambda }$ corresponding to the end of the first stage. In Ref. [14] for MOCVD growth and Ref. [26] for the MBE growth of Au-catalyzed III-V NWs, exponential NW growth was considered, in which the NW length scales exponentially with H or t. This exponential stage is more relevant for NW growth on unpatterned substrates [26,28], while Equation (8) predicts a more complex behavior. It is reduced to the exponential growth law only when $4\chi_{nw}H/R_{0}\ll \ln\left[\left(2-a\right)/a\right)]≅-lna$:(9)$$L=\frac{\chi_{d}}{\chi_{nw}}\frac{R_{0}}{\left(1+cos\beta \right)}\left[exp\left(\frac{4\chi_{nw}H}{R_{0}}\right)-1\right].\;$$

In dilute NW arrays with ${\tilde{S}}_{nw}<cP^{2}$, the first stage continues as long as NWs are shorter than the diffusion length of group III adatoms on the NW sidewalls $\lambda_{3}^{des}$. This $\lambda_{3}^{des}$ is shorter than the critical length $L_{*}$ corresponding to $S_{nw}\left(\;L_{*}\right)=cP^{2}$. The critical length is given by
(10)$$L_{*}=\frac{cP^{2}}{\chi_{nw}2\pi R_{0}}-\frac{\chi_{d}}{\chi_{nw}}\frac{R_{0}}{\left(1+cos\beta \right)}=R_{0}\frac{\chi_{d}}{\chi_{nw}}\frac{R_{0}}{\left(1+cos\beta \right)}\left(\frac{1}{a}-1\right)$$

The linear growth of NWs in dilute arrays in the second stage is easily obtained from Equations (1) and (5) at $R=R_{0}$ and β=const:(11)$$L=\lambda_{3}^{des}+\left(\frac{2\chi_{d}}{1+cos\beta }+\chi_{nw}\frac{2\lambda_{3}^{des}}{R_{0}}\right)\left(2-b\right)\left(H-H_{\lambda }\right),\;H>H_{\lambda },\;b=a+\frac{\chi_{nw}2\pi R_{0}\lambda_{3}^{des}}{cP^{2}}.\;$$

In dense NW arrays with $\lambda_{3}^{des}>L_{*}$, linear NW growth law in the second stage is obtained from Equations (1) and (6) in the form
(12)$$L=L_{*}+\frac{cP^{2}}{\pi R_{0}^{2}}\left(H-H_{*}\right),\;$$
similar to Refs. [29,32,33,34]. It is interesting to note that the three growth modes yield different radius correlations of the NW growth rate or N—inverse exponential dependence on $R_{0}$ given by Equation (9), inverse radius dependence given by Equation (11) and $1/R_{0}^{2}$ dependence given by Equation (12). Such separation of MOCVD growth regimes was discussed earlier, for example, in Refs. [14,20,21]. However, there are some important differences. First, including the additional flux of group III atoms originating from desorbed species results in the growth law given by Equation (8). This law is more complex than exponential and depends on pitch P. It is reduced to the exponential regime only for very large pitches (a→0), and the factor $4\chi_{nw}H/R_{0}$ in the exponential of Equations (8) and (9) is two-fold larger than in Ref. [14] due to the re-emission of group III atoms from the surface. Second, linear growth law in dilute arrays of NWs given by Equation (11) contains the pitch-dependent correction b. Third, the $1/R_{0}^{2}$ radius dependence of the NW length in Equation (12) originates from the saturation of the total material current into the NW rather than from pitch-dependent surface diffusion from the substrate surface [20]. These differences will be further discussed in the next section. 

Figure 1 shows the NW length as a function of deposition thickness, obtained from Equations (8), (11) and (12) for a fixed NW radius $R_{0}$ of 30 nm, diffusion length $\lambda_{3}^{des}$ of 1500 nm, droplet contact angle *β* of 130°, *c* = 0.3, $\chi_{d}$ = 1, $\chi_{nw}$ = 0.5, and variable pitches P from 200 nm to 1000 nm. According to Table 1, the NW growth kinetics is limited by pitch for P= 200 nm, 400 nm, and 600 nm because the critical length $L_{*}$ is shorter than 1500 nm. For larger pitches of 800 nm and 1000 nm, the NW growth kinetics is limited by the desorption of group III adatoms from the NW sidewalls. In any case, non-linear evolution of the NW length with H in the first stage is followed by linear axial growth on the second stage. For a given deposition thickness or growth time, NWs are longer for larger pitches. 

Figure 2 shows the NW length as a function of deposition thickness, obtained from Equations (8), (11), and (12) for the same parameters as in Figure 1 at a fixed pitch of 800 nm and different NW radii $R_{0}$ from 15 to 75 nm. According to Table 2, the NW growth kinetics is limited by pitch for thicker NWs with 45 nm, 60 nm, and 75 nm radii. Thinner NWs grow much faster, and the conversion from non-linear growth in stage 1 to linear growth in stage 2 occurs at $L=\lambda_{3}^{des}=$ 1500 nm. 

Simultaneous axial and radial growth kinetics of III-V NWs is much more complex than axial growth at a constant radius. The asymptotic growth stage in dense arrays of NWs with $L>L_{*}$ from the maximum material current given by Equation (6) has been described in Ref. [32] and is very similar in MBE and MOCVD techniques. Some MOCVD-grown III-V NWs extend radially from the very early stage, particularly in the case of catalyst-free SAE [17]. Treating the radial NW growth requires the introduction of another diffusion length of group III adatoms on the NW sidewalls, $\lambda_{3}^{inc}$, which is limited by surface incorporation rather than desorption. Clearly, $\lambda_{3}^{inc}<\lambda_{3}^{des}$, because no radial growth may occur otherwise. The elongation of short NWs with $L<\lambda_{3}^{inc}$ at an initial radius $R_{0}$ is given by Equation (8). For $\lambda_{3}^{inc}<L<\lambda_{3}^{des}$, we have
(13)$$\begin{array}{l} \frac{d}{dH}\left(\pi R^{2}L\right)=\left(\chi_{d}\frac{2\pi R^{2}}{1+cos\beta }+\chi_{nw}2\pi RL\right)\left(2-\frac{S_{nw}}{cP^{2}}\right),\;\\ \frac{dL}{dH}=\left(\frac{2\chi_{d}}{1+cos\beta }+\frac{2\chi_{nw}\lambda_{3}^{inc}}{R}\right)\left(2-\frac{S_{nw}}{cP^{2}}\right). \end{array}$$

Here, the first equation follows from Equations (1) and (2) at β=const. The second equation means that the axial NW growth is due to the collection of the group III atoms by the droplet (or NW top facet for catalyst-free SAE NWs) and the upper NW section of length $\lambda_{3}^{inc}$, with the contribution of group III atoms desorbed from the substrate surface. From Equation (13), the radial NW growth rate is given by
(14)$$\frac{dR}{dH}=\left(1-\frac{\lambda_{3}^{inc}}{L}\right)\left(2-\frac{S_{nw}}{cP^{2}}\right).\;$$

After dividing Equation (14) by Equation (13) and integrating, one obtains
(15)$$\frac{2\chi_{d}}{1+cos\beta }\left(R-R_{0}\right)+2\chi_{nw}\lambda_{3}^{inc}ln\left(\frac{R}{R_{0}}\right)=L-\lambda_{3}^{inc}-\lambda_{3}^{inc}ln\left(\frac{L}{\lambda_{3}^{inc}}\right).\;$$

This solution gives the NW radius as a function of its length, while the NW length versus H should be then obtained by numerical integration of Equation (13). Ignoring the droplet surface or NW top surface for SAE NWs, and assuming $\chi_{nw}=1$, the NW volume scales exponentially with its length
(16)$$\pi R^{2}L=\pi R_{0}^{2}\lambda_{3}^{inc}exp\left(\frac{L}{\lambda_{3}^{inc}}-1\right),\;$$
similar to Ref. [32]. The case of $\chi_{nw}=1$ should be a good approximation for high temperature MOCVD growth where all group precursors crack on the NW sidewalls. 

For $L>\lambda_{3}^{des}$, in the case of $\lambda_{3}^{des}<L_{*}$, the NW growth kinetics is given by
(17)$$\begin{array}{l} \frac{d}{dH}\left(\pi R^{2}L\right)=\left(\chi_{d}\frac{2\pi R^{2}}{1+cos\beta }+\chi_{nw}2\pi R\lambda_{3}^{des}\right)\left(2-\frac{{\tilde{S}}_{nw}}{cP^{2}}\right),\;\\ \frac{dL}{dH}=\left(\frac{2\chi_{d}}{1+cos\beta }+\frac{2\chi_{nw}\lambda_{3}^{inc}}{R}\right)\left(2-\frac{{\tilde{S}}_{nw}}{cP^{2}}\right). \end{array}$$

Here, the first equation follows from Equations (1) and (5) at β=const. The second equation is similar to Equation (13). Using these equations, the radial NW growth rate is given by
(18)$$\frac{dR}{dH}=\frac{\left(\lambda_{3}^{des}-\lambda_{3}^{inc}\right)}{L}\left(2-\frac{{\tilde{S}}_{nw}}{cP^{2}}\right),\;$$
which is different from Equation (14) due to a limited collection length of group III adatoms on the NW sidewalls. From Equations (17) and (18), it is easy to obtain
(19)$$\frac{2\chi_{d}}{1+cos\beta }\left(R-R_{1}\right)+2\chi_{nw}\lambda_{3}^{inc}ln\left(\frac{R}{R_{1}}\right)=\left(\lambda_{3}^{des}-\lambda_{3}^{inc}\right)ln\left(\frac{L}{\lambda_{3}^{des}}\right).\;$$

Ignoring, as above, the droplet surface or NW top surface, and, assuming $\chi_{nw}=1$, one arrives at
(20)$$\pi R^{2}L=\pi R_{1}^{2}\lambda_{3}^{des}{\left(\frac{L}{\lambda_{3}^{des}}\right)}^{\lambda_{3}^{des}/\lambda_{3}^{inc}}.\;$$

Therefore, NW volume features an exponential increase with its length in the absence of the desorption of group III atoms from the NW sidewalls, and only power-law increases in the presence of desorption. From Equation (20), the NW radius increases as a power-law of its length: (21)$$R=R_{1}{\left(\frac{L}{\lambda_{3}^{des}}\right)}^{\alpha },\;\alpha =\frac{1}{2}\left(\frac{\lambda_{3}^{des}}{\lambda_{3}^{inc}}-1\right),\;$$
with α>0. For very low-density arrays of NWs, the ${\tilde{S}}_{nw}/cP^{2}$ terms in Equations (17) and (18) can be neglected. In this case, the NW length and radius versus deposition thickness are given by
(22)$$L=\lambda_{3}^{des}{\left[1+\frac{4\left(\alpha +1\right)}{2\alpha +1}\frac{\left(H-H_{1}\right)}{R_{1}}\right]}^{1/\left(\alpha +1\right)}\;,\;R=R_{1}{\left[1+\frac{4\left(\alpha +1\right)}{2\alpha +1}\frac{\left(H-H_{1}\right)}{R_{1}}\right]}^{\alpha /\left(\alpha +1\right)}.\;$$

Therefore, the length and radius of low-density NWs in this stage increase sub-linearly with H or t. This result is similar to the models of Ref. [23] for catalyst-free SAE III-V NWs and Ref. [38] for catalyst-free self-induced GaN NWs grown on unpatterned SiN_x_/Si(111) substrates, where no pitch-dependent cooperative effects of NW growth were included. 

## 4. Theory and Experiment

The superlinear increase in the NW length with time in the initial growth stage, followed by linear or even sublinear time evolution in a later stage, has been experimentally demonstrated for the MBE growth of Au-catalyzed InP_1−x_As_x_ [26] and GaAs [39] NWs on unpatterned substrates, for MOCVD growth of Au-catalyzed InAs NWs [14,20], and for catalyst-free SAE GaAs NWs [17] on patterned substrates. Superlinear exponential stage of III-V NW growth has previously been explained by the contribution of the surface diffusion of group III adatoms from the NW sidewalls to the top [14,20,26,39], which is proportional to the NW length L. The radial NW growth significantly complicates the NW growth kinetics [17,29,39]. Below, I consider the data on the MOCVD growth of III-V NWs on patterned substrates, assuming the pitch-dependent re-emission of group III atoms from an inert mask surface as the main mechanism of material exchange between the substrate and NWs. 

InAs NWs of Refs. [14,20] were grown by A-catalyzed MOCVD on InAs(111)B substrates at a temperature of 450 °C. Before MOCVD growth, regular hexagonal ($k=\sqrt{3}/2$) arrays of Au nanoparticles were prepared by means of electron beam lithography (EBL), followed by the thermal evaporation of Au and lift-off. The distance between Au nanoparticles (pitch P) was varied from 150 nm to 1000 nm. By varying the EBL dose, the pinhole radius was varied between 10 and 50 nm. The measured NW radii varied between 10 and 55 nm, and no significant radial growth was observed [14,20]. The investigated growth times were 7.5, 15, 22.5, 30, and 60 min.

Figure 3a shows the time evolution for the length of 50–55 nm radius InAs NWs [14], with a solid black line representing short NWs and a dashed line representing long NWs which correspond to the fit of Ref. [14], obtained from the model for individual NW growth kinetics given by
(23)$$\frac{dL}{dt}=v_{3}\left(\frac{2\chi_{d}}{1+cos\beta }+2\chi_{nw}\frac{L}{R_{0}}\right),\;L\leq \lambda_{3}^{des},\;\;\frac{dL}{dt}=C_{fit}v_{3}\left(\frac{2\chi_{d}}{1+cos\beta }+2\chi_{nw}\frac{\lambda_{3}^{des}}{R_{0}}\right),\;L>\lambda_{3}^{des}.$$

Clearly, these equations give exponential growth law for short NWs with $L\leq \lambda_{3}^{des}$ and linear growth law for long NWs with $L>\lambda_{3}^{des}$. At a constant NW radius $R_{0}=$ 52.5 nm, the fitting parameters are: $\chi_{d}/\left[\chi_{nw}\left(1+cos\beta \right)\right]=$ 1.51, $2\chi_{nw}v_{3}=$ 7.89, $\lambda_{3}^{des}=$ 1450 nm, and $C_{fit}=$ 0.247. The exponential character of MOCVD growth in the first stage is supported by the broad Polya-like length distributions of NWs whose variance scales linearly with the mean length [14]. This model provides a good fit to all datapoints, with the transition from exponential to linear time dependence of the NW length at a desorption-limited diffusion length of Ga adatoms on the NW sidewalls of 1450 nm. However, the obtained correspondence requires the introduction of the fitting coefficient $C_{fit}=$ 0.247 in Equation (23). This choice was not justified in Ref. [14] and leads to the discontinuity of the NW growth rate at $L=\lambda_{3}^{des}$, as shown in Figure 3b. There is no physical reason for such an abrupt (more than 4 times) decrease in the axial growth rate at the threshold, apart from the onset of radial growth or parasitic surface growth, which were not observed in Refs. [14,20]. Furthermore, the axial NW growth rate should remain continuous even if the Ga diffusion length is limited by surface incorporation according to Refs. [32,33,34]. I speculate that this unphysical discontinuity is due to the absence of any pitch dependence in Equation (23). For the continuous axial growth rate at $L=\lambda_{3}^{des}$ corresponding to $C_{fit}=$ 1, the growth model for individual NWs gives a largely overestimated NW length for longer times (t> 20 min), as shown by the solid black curve in Figure 3a. 

The fit obtained from Equations (8) and (11) at $\chi_{d}/\left[\chi_{nw}\left(1+cos\beta \right)\right]=$ 1.51, $2\chi_{nw}v_{3}=$ 4.3, $\lambda_{3}^{des}$ = 1400 nm, and $\chi_{s}=1$, corresponding to c=k, $F_{max}=v_{3}kP^{2}$, and a=0.022, is shown by the blue curve in Figure 3a. It is almost indistinguishable from the previous result. The curve fits the datapoints quite well for short NWs in the quasi-exponential growth stage, remains continuous at $L=\lambda_{3}^{des}$, but largely overestimates the NW length for longer times. This is not surprising because the small a yields an almost exponential growth of short-enough NWs given by Equation (9). The fitting value of $2\chi_{nw}v_{3}=$ 4.3 is almost two-fold lower than $2\chi_{nw}v_{3}=$ 7.89 used for the fit by Equation (23), because the axial growth rate in our model is twice as large due to re-emitted In flow. The transition from exponential to linear growth regime is limited by In desorption from the NW sidewalls because the critical length $L_{*}=$ 3525 nm is larger than $\lambda_{3}^{des}$. 

At $\chi_{s}=1$, the maximum group III current into the NW, $F_{max}=v_{3}kP^{2}$, corresponds to situation where all group III atoms arriving from vapor are collected by NWs. Such a growth regime is typical for MBE [29,32], though is hardly possible in MOCVD. Therefore, I use $\chi_{s}$ as a free parameter to obtain the continuous axial NW growth rate, fitting the datapoints in Figure 3a in the entire range of the investigated growth times. The red curve in Figure 3a is the best fit obtained using Equations (8) and (12) at $\chi_{d}/\left[\chi_{nw}\left(1+cos\beta \right)\right]=$ 3.15, $2\chi_{nw}v_{3}=$ 3.7, and $\chi_{s}=$ 0.143. This $\chi_{s}$ corresponds to c=0.124, a larger a of 0.14 and a smaller critical length $L_{*}$ of 1020 nm, which is shorter than the desorption-limited diffusion length of In adatoms. Assuming $\chi_{d}=1$ (100% precursor decomposition at the droplet surface) and β= 90°, the cracking efficiency of In precursor at the NW sidewalls $\chi_{nw}$ equals 0.317. With these parameters, the transition from non-linear to linear axial growth is limited by the saturation of the total In current collected by NWs at $L=L_{*}$ rather than In desorption. The curve fits all the datapoints in Figure 3a, and the continuous axial growth rate is shown in Figure 3b. 

The model is further supported by the pitch-dependent growth kinetics of 26–27.5 radius InAs NWs shown in Figure 4a [20]. The effect of pitch is very substantial. In particular, the NWs grown for 60 min in 200 pitch arrays are shorter than 3000 nm, while NWs grown in 500 nm pitch arrays are longer than 7000 nm. All datapoints are well fitted by Equations (8) and (12) at a fixed $2\chi_{nw}v_{3}=$ 3.7, $\chi_{d}/\left[\chi_{nw}\left(1+cos\beta \right)\right]=$ 3.15, and other parameters are summarized in Table 3. The fitting values of the effective adsorption coefficient $\chi_{s}$ are very close for all pitches (from 0.21 to 0.29) except the smallest pitch of 200 nm, where $\chi_{s}$ equals 0.512. Gradual decrease in $\chi_{s}$ with an increasing pitch is observed. This effect is probably related to a larger contribution of group III atoms desorbed from the NW sidewalls or other effects which require a separate study and will be considered elsewhere. In any case, the model explains the pitch dependence of the NW length and fits the data with plausible parameters. Most importantly, it eliminates the non-physical discontinuity of the NW growth rate in the transition from exponential to linear growth [14].

Figure 4b shows the length–diameter correlation of InAs NWs grown for 30 min in 300 nm pitch array. Since the critical length $L_{*}$ for this sample is only 337 nm according to Table 3, one may expect the $1/R_{0}^{2}$ radius dependence of the NW length following from Equation (12). The fitting curve shown in Figure 4b is obtained from Equation (12), with $cP^{2}=$ 22,470 nm^2^, according to Table 3, and fits very well the data. The pitch dependence of InAs NW length and the $1/R_{0}^{2}$ radius correlation was earlier explained in Ref. [20] by In diffusion flux from the mask surface with a competition for this flux between the neighboring NWs. This explanation cannot be entirely ruled out. However, NWs with lengths larger than ~1400 nm should not be affected by the In diffusion flux from the substrate because all In atoms should evaporate before reaching the NW top. Conversely, Figure 4 shows the same linear increase in length with time and $1/R_{0}^{2}$ radius correlation for much longer NWs. 

GaAs NWs of Ref. [17] were grown by MOCVD at 750 °C using catalyst-free SAE on lithographically patterned SiO_2_/GaAs substrates, with a 600 nm pitch and variable pinhole diameter from 125 to 225 nm. These NWs had an almost uniform diameter from base to top, without any pronounced tapering. Uniform diameter was probably maintained by step flow on the NW sidewalls, as described in Refs. [29,32,33,34,40] for VLS NWs. No parasitic growth was observed between the NWs. The radial growth of these NWs started from the very early stage, suggesting that the incorporation-limited diffusion length of Ga adatoms on the NW sidewalls, $\lambda_{3}^{inc}$, is very short. On the other hand, one may expect a short desorption-limited diffusion length $\lambda_{3}^{des}$ at a high growth temperature of 750 °C. According to our analysis of the radial NW growth, the volume of short NWs with $\lambda_{3}^{inc}<L<\lambda_{3}^{des}$ increases exponentially with L according to Equation (16), while the volume of longer NWs with $L>\lambda_{3}^{des}$ increases as a power-law of L according to Equation (20). 

Figure 5 shows the average NW volume versus their average length for three different sizes of pores. This increase is not exponential but can be fitted with the power-law dependence given by Equation (20), with the parameters summarized in Table 4. This power-law behavior is not influenced by pitch according to our analysis. The values of $\lambda_{3}^{inc}$ obtained from the best fit for each growth experiment are in the range of ~300–400 nm, while the values of $\lambda_{3}^{des}/\lambda_{3}^{inc}$ are in the order of 2. Hence, Ga desorption starts from ~800 nm length, while the measured NW lengths extend to 3000 nm. This explains why the power-law works quite well from the very beginning of growth. Conversely, the exponential fit can only be used in the very beginning of growth (for short NWs with lengths shorter than 1000 nm); however, it does not work for longer NWs. 

## 5. Conclusions

To summarize, a new approach has been developed for modeling the growth kinetics of III-V NWs obtained by MOCVD technique in regular arrays on patterned substrates. The main difference with the existing models for MOCVD NW growth is in the neglect of the surface diffusion of group III atoms on an inert mask and considering the additional flux of re-emitted group III atoms instead. The model predicts a continuous NW growth rate which saturates either due to the desorption of group III atoms from the NW sidewalls or reaching a pitch-dependent maximum group III current which NWs can collect. The model fits quite well the data on the axial growth of Au-catalyzed InAs NWs, with continuous axial growth rate at the transition from non-linear to linear growth and catalyst-free SAE GaAs NWs. The developed approach is quite general and should be useful for understanding and tuning the morphology of NWs in different material systems such as elemental semiconductors, III-nitrides, and oxides. It can also be extended to other epitaxy techniques, for example, hydride VPE, which enables the fast synthesis of III-V NWs with exceptional length [41]. I now plan to refine some assumptions of the model and consider in more detail the limits of NW growth in MOCVD techniques, the pitch dependence of the NW morphology, the radial NW growth and possible ways to suppress it, and the statistical properties within the ensembles of MOCVD-grown NWs. 

## Figures and Tables

**Figure 1 nanomaterials-12-02632-f001:**
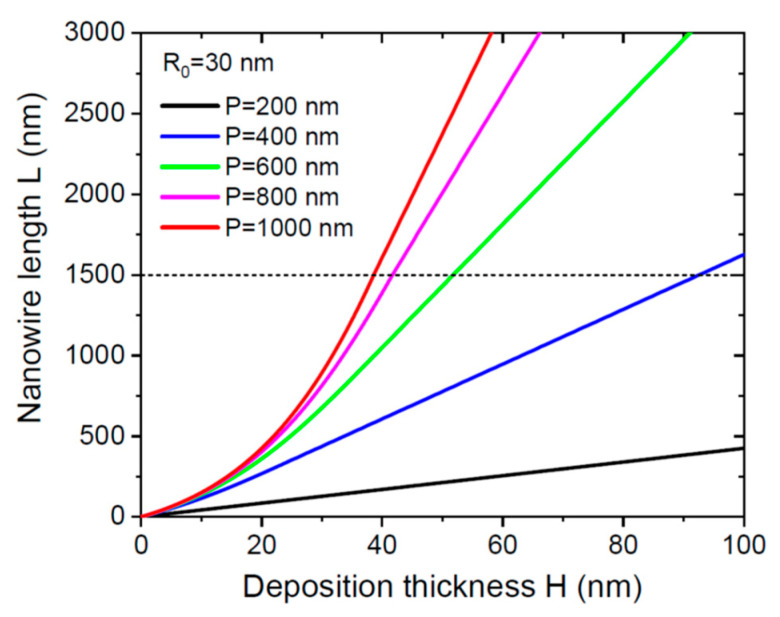
NW length versus deposition thickness at a fixed NW radius of 30 nm for different pitches shown in the legend. The dashed line corresponds to a diffusion length $\lambda_{3}^{des}$ of 1500 nm.

**Figure 2 nanomaterials-12-02632-f002:**
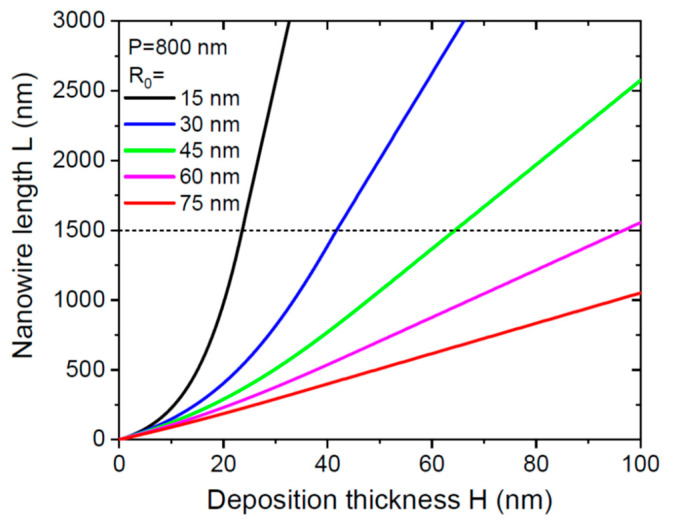
NW length versus deposition thickness at a fixed pitch of 800 nm for different NW radii shown in the legend. The dashed line corresponds to a diffusion length $\lambda_{3}^{des}$ of 1500 nm.

**Figure 3 nanomaterials-12-02632-f003:**
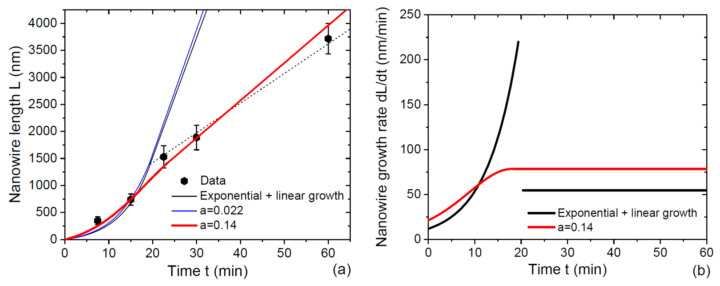
(**a**) Length of 50–55 nm radius Au-catalyzed InAs NWs grown by Au-catalyzed MOCVD at 450 °C in 1000 nm pitch arrays versus time (symbols) [14], fitted by different models at a constant NW radius $R_{0}=$ 52.5 nm (lines). Solid black line shows the fit by Equation (23) at $C_{fit}=1,$ where the NW axial growth rate at $L=\lambda_{3}^{des}=$ 1450 nm is continuous. The dashed line is the fit of Ref. [14] at $C_{fit}=$ 0.247, with discontinuous NW growth rate at $L=\lambda_{3}^{des}.$ The blue line is the fit by Equations (8) and (11) at $\chi_{d}/\left[\chi_{nw}\left(1+cos\beta \right)\right]=$ 1.51, $2\chi_{nw}v_{3}=$ 4.3, $\lambda_{3}^{des}=$ 1400 nm, and $\chi_{s}=1$, corresponding to a=0.022. The red line is the fit obtained from Equations (8) and (12) with $\chi_{d}/\left[\chi_{nw}\left(1+cos\beta \right)\right]=$ 3.15, $2\chi_{nw}v_{3}=$ 3.7, and $\chi_{s}=$ 0.143 (corresponding to c=0.124 and a=0.14 ), with continuous NW growth rate at $L=L_{*}=$ 1020 nm. (**b**) Discontinuous NW growth rate was used in Ref. [14] (the black line) and the continuous NW growth rate was obtained in this work (the red line).

**Figure 4 nanomaterials-12-02632-f004:**
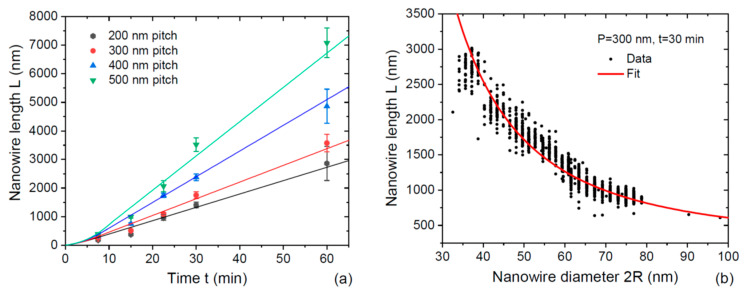
(**a**) Length of 26–27.5 nm radius InAs NWs versus time for different pitches P shown in the legend (symbols) [20], fitted by Equations (8) and (12), with the parameters summarized in Table 3 (lines). (**b**) Length–diameter correlation of InAs NWs grown for 30 min in 300 nm pitch array (symbols) [20], fitted by the model (line).

**Figure 5 nanomaterials-12-02632-f005:**
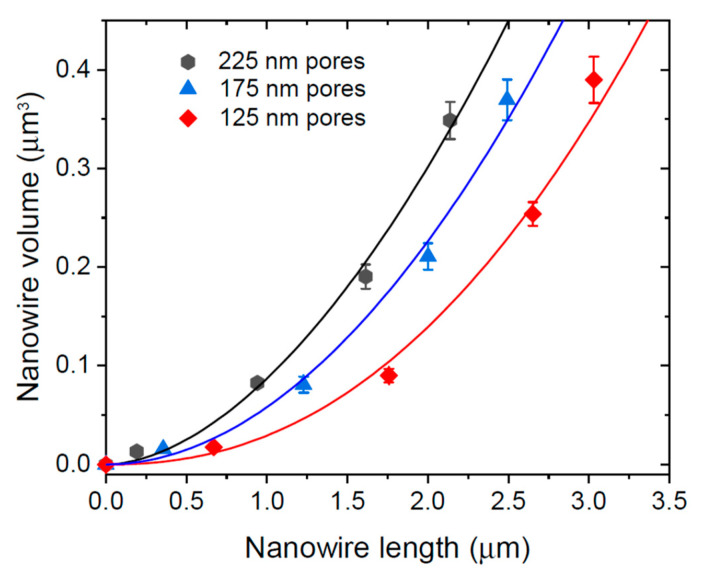
Volume of GaAs NWs grown by catalyst-free SAE MOCVD at 750 °C in regular arrays of patterned pores in SiO_2_/GaAs versus the NW length for three different pore diameters shown in the legend (lines), fitted by Equation (20) with the parameters summarized in Table 4 (lines).

**Table 1 nanomaterials-12-02632-t001:** Parameters of pitch-dependent NW growth kinetics.

Pitch P (nm)	*a*	$\mathrm{Critical}\;\mathrm{Length}\;L_{*}\;(\mathrm{nm})$	b	$cP^{2}/\left(\pi R_{0}^{2}\right)$
200	1.319	0	-	4.25
400	0.330	341	-	16.99
600	0.147	975	-	38.2
800	0.0823	1873	0.818	-
1000	0.0527	3020	0.524	-

**Table 2 nanomaterials-12-02632-t002:** Parameters of radius-dependent NW growth kinetics.

$\mathrm{Radius}\;R_{0}\;(\mathrm{nm})$	*a*	$\mathrm{Critical}\;\mathrm{Length}\;L_{*}\;(\mathrm{nm})$	b	$cP^{2}/\left(\pi R_{0}^{2}\right)$
15	0.0210	3916	0.368	-
30	0.0823	1873	0.818	-
45	0.186	1103	-	30.2
60	0.329	685	-	16.99
75	0.514	397	-	10.87

**Table 3 nanomaterials-12-02632-t003:** Parameters of pitch-dependent growth kinetics of Au-catalyzed InAs NWs.

Pitch P (nm)	a	$\mathrm{Critical}\;\mathrm{Length}\;L_{*}\;(\mathrm{nm})$	$\chi_{s}$	$cP^{2}\;({\mathrm{nm}}^{2})$
200	0.25	253	0.512	17975
300	0.20	337	0.288	22470
400	0.12	564	0.249	34595
500	0.09	852	0.213	46130

**Table 4 nanomaterials-12-02632-t004:** Parameters of radius-dependent growth kinetics of SAE GaAs NWs.

Pore Diameter (nm)	Average initial NW Diameter (nm)	$\lambda_{3}^{inc}\;(\mathrm{nm})$	$\lambda_{3}^{des}/\lambda_{3}^{inc}$
225	228	300	1.79
175	173	320	1.96
125	122	410	2.25

## Data Availability

Not applicable.

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
