# Peer review of "Theory of MOCVD Growth of III-V Nanowires on Patterned Substrates"

_nanomaterials, 2022, doi:10.3390/nano12152632_

Round 1
Reviewer 1 Report
1.The Snw represents the effective area of effective surface area of the Snw, what is the physical meaning of ~Snw(Equation (2).
2. When studying the relationship between the length of nanowires and the growth time, the experimental data in one literature is a bit small. Can you give more experimental data on the growth of III-V nanowires in other literatures to support the theoretical model.
3. In Figure 3, the linear and exponential regions give too few datapoints for a data fit. It will be more convincing if you can provide more datapoints.
Author Response
Please see the response letter attached.

Reviewer 2 Report
The paper presents a theoretical model for MOCVD growth of III-V nanowires (NWs). Although a deeply discussed topic, the Author suggested a different approach for group III atoms, i.e., neglecting their surface diffusion and considering only re-emission. Different scenarios are considered, with equations explaining NWs features in terms of growth parameters. The comparison with some experimental data is quite good, able to explain or, at least, better define the growth details and characteristics. The paper is well written and promises further developments, considering other NWs growth parameters. Concerning typos, in line 351 “form” should be “from”.
Author Response

(The authors gave the same response as above.)
